# COMPUTE OR LOAD KV CACHE? WHY NOT BOTH?

## ABSTRACT

Recent advancements in Large Language Models (LLMs) have significantly increased context window sizes, enabling sophisticated applications but also introducing substantial computational overheads, particularly computing key-value (KV) cache in the prefill stage. Prefix caching has emerged to save GPU power in this scenario, which saves KV cache at disks and reuse them across multiple queries. However, traditional prefix caching mechanisms often suffer from substantial latency because the speed of loading KV cache from disks to GPU memory is bottlenecked by the throughput of I/O devices. To optimize the latency of long-context prefill, we propose *Cake*, a novel KV cache loader, which employs a bidirectional parallelized KV cache generation strategy. Upon receiving a prefill task, *Cake* simultaneously and dynamically loads saved KV cache from prefix cache locations and computes KV cache on local GPUs, maximizing the utilization of available computation and I/O bandwidth resources. Additionally, *Cake* automatically adapts to diverse system statuses without manual parameter. tuning. In experiments on various prompt datasets, GPUs, and I/O devices, *Cake* offers up to 68.1% Time To First Token (TTFT) reduction compare with compute-only method and 94.6% TTFT reduction compare with I/O-only method.

## 1 INTRODUCTION

Large Language Models (LLMs) have been widely adopted in large-scale online services, making efficient online serving of these models a critical research and engineering challenge (Kwon et al., 2023; Agrawal et al., 2024; Zheng et al., 2023; Miao et al., 2024; Leviathan et al., 2023; Ning et al., 2023; Jin et al., 2024b). Recent advancements in LLM development have significantly expanded the models' context windows, enabling sophisticated applications such as long document understanding (Wang et al., 2024), long-context Retrieval Augmented Generation (RAG) (Jiang et al., 2024), and complex LLM agents (Zhang et al., 2024). For instance, GPT-4 boasts a context window of 128k tokens (openAI, 2024), while Claude-3.5 Sounet extends this further to 200K tokens (Anthropic, 2024). However, processing these long-context prompts introduces substantial computational overhead, particularly in the prefill stage, where the key-value (KV) cache is calculated. [1] For example, generating KV cache for a 200-page book like "The Great Gatsby" (approximately 72K tokens) requires about 180GB of memory. For a 70B parameter model, generating such a KV cache on an A100 GPU takes approximately 30 seconds, significantly impacting user experience.

To mitigate this overhead, prefix caching, i.e., the cache of KV cache, has emerged as a useful mechanism. This approach is particularly effective in applications where long contexts are frequently and repeatedly used across multiple requests. For example, in long document processing tasks, the KV cache of a large document can be reused for multiple queries about its content. Similarly, in coding assistance scenarios, a cached summary of the codebase can be reused across multiple code completion or Q&A requests. LLM service providers like Claude and Deepseek have begun implementing such prefix caching mechanism in their online services (Anthropic, 2023; Deepseek, 2024). Recent studies have proposed various system solutions to implement prefix caching (Juravsky et al., 2024; Jin et al., 2024a; Gim et al., 2024). LLM inference engines such as vLLM (Kwon et al., 2023) and SGLang (Zheng et al., 2023) cache KV in local CPU memory, but this approach is limited by CPU memory size. Given that disks offer much larger capacity and are more cost-efficient than CPU

---

[1]KV cache is an essential technique to reduce the computational overhead of LLM inference in the decoding stage for each request and widely adopted in the state-of-the-art inference system.

memory, storing KV caches in local or remote disks has become a more viable option (Gao et al., 2024; AutoGen, 2024). CacheGen (Liu et al., 2023) proposes solutions to optimize KV streaming from local or remote disks to GPU memory, aiming for scalable prefix caching.

In practice, long-context LLM inference workflow with prefix caching is depicted in Figure 1. Upon receiving an LLM request, the serving system first checks whether it has a prefix with available corresponding KV cache. If so, the system directly loads KV cache from the cache location to the GPU's memory, saving the overhead of recomputation. Note that prefix caching may involve multiple levels, including CPU memory, local disks, and remote disks. After loading the cached prefix, the system continues the prefill of remaining tokens in the request and proceeds with subsequent steps.

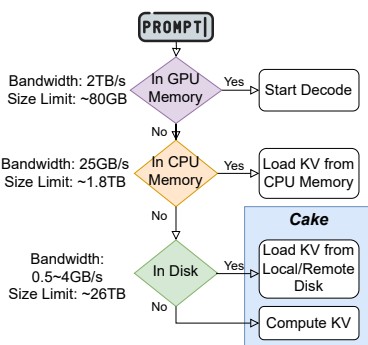

Figure 1: Workflow of long-context LLM inference with prefix caching. *Cake* operates in the KV loading phase (highlighted in blue). The configuration parameter is based on the specification of LambdaLab Server (Lambda, 2024).

However, while prefix caching saves GPU computation, it is not optimized for latency, specifically Time-to-First-Token (TTFT), one of the key metrics in LLM serving. When the prefix cache is not found in local CPU memory (a common scenario due to CPU memory constraints), streaming KV cache data from disk or network is often not faster than recomputing KV cache if GPU power is available as we demonstrated in § 3. Additionally, both the available computation and I/O resources vary based on the system workload. Consequently, the two approaches of obtaining KV cache-GPU computation and prefix cache loading - are only beneficial in terms of latency when there is sufficient GPU power or I/O bandwidth, respectively. Currently, there is a lack of dynamic KV cache management systems that can optimize LLM inference latency across diverse system statuses.

To address this gap, we propose *Cake* (Computation and Network Aware KV CachE loader), a system designed to achieve optimal latency for long-context LLM inference. *Cake* simultaneously leverages local computation and data streaming from prefix cache locations to generate the KV cache. It employs a bidirectional parallelized KV cache generation strategy: upon receiving a prefill task, *Cake* immediately utilizes available GPU power to compute KV cache in normal order starting from the first token, while simultaneously loading the prefix cache in reverse order starting from the last token, until all tokens have corresponding KV cache data. This design is inspired by key observations about the characteristics of computing and streaming KV cache: the computational cost of generating KV cache increases with the token's distance from the beginning of the sequence, while the data streaming cost remains constant regardless of token position. Importantly, *Cake*'s design is both simple and elegant, adapting to diverse situations in terms of system architecture and available resources, without requiring manual parameter tuning.

The main contribution of this paper includes the design and evaluation of *Cake*, which enables the following features improving the quality of LLM serving.

• **Optimized latency:** *Cake* optimizes the overall latency of KV cache loading by achieving maximum utilization of computation and network streaming in parallel without any idle time.

• **Automatic adaptation:** *Cake* continuously adapts to the available network and computational resources, achieving low latency without downgrading system performance. Previous computation-only and network-only KV cache prefill solutions are special cases of *Cake*.

• **Negligible Overhead:** *Cake* introduces minimal computational and memory overhead, ensuring it doesn't negatively impact the performance of the underlying system.

Our experiments demonstrate that *Cake* efficiently utilizes both computation and I/O fetching to significantly reduce the prefilling latency in long-context LLM inference. Evaluations on diverse datasets, including LongChat (Li et al., 2023a), TriviaQA, and NarrativeQA (Bai et al., 2023), show that *Cake* achieves substantial improvements in Time-to-First-Token (TTFT). Compared to compute-only methods, *Cake* achieves an average 36.7% reduction in TTFT. When compared to

I/O-only methods, the improvement is even more pronounced, with an average 60.55% reduction in TTFT. Notably, *Cake* accomplishes these performance gains while introducing minimal overhead to the system, making it a highly efficient and practical solution for optimizing long-context LLM inference across various scenarios and workloads.

## 2 BACKGROUND

### 2.1 KV CACHE IN LLM INFERENCE

Large language models (LLMs) have recently made significant impacts across numerous domains. The key to their success lies in the attention mechanism, which enables these models to scale up and effectively process long contexts (Vaswani et al., 2017; Brown et al., 2020). In the attention calculation process, the computation of Key (K) and Value (V) vectors for previously processed tokens becomes redundant during the decoding phase. Recognizing this, the concept of KV cache (Zhang et al., 2023) is introduced. This approach involves storing these computed values and reusing them to reduce computational overhead in subsequent decoding steps.

State-of-the-art LLM inference engines (Kwon et al., 2023; Miao et al., 2024; Zheng et al., 2023) typically divide the inference procedure into two distinct phases: prefill and decode. The prefill stage generates the initial KV cache for the input prompt. In the subsequent decode stage, the model utilizes this cache to generate new tokens sequentially. As each new token is produced, its corresponding K and V vectors are computed and appended to the KV cache. This caching mechanism significantly accelerates inference by converting the time complexity of token generation from quadratic to linear (Yang et al., 2024).

### 2.2 KV CACHE SAVING AND REUSING

The prefill stage is a highly resource-consuming procedure that often saturates GPU computation resources and causes high latency compared to the normal decoding procedure (Agrawal et al., 2024). This overhead becomes even more significant in long context scenarios (Fu, 2024).

Meanwhile, it is common for parts of prompts to be reused across multiple requests. For instance, system messages in chatbot services guiding the LLM model's behavior are usually long and shared across multiple messages. In Retrieval-Augmented Generation scenarios, fetching long text blocks as the context to generate answers can improve the generation performance (Jiang et al., 2024). Other use cases include coding assistants that need to maintain a summarized version of the codebase in the prompt (Cursor, 2024), and agentic search, tool use and multi-agent communication which require multiple rounds of API calls using the same set of historical data (Wu et al., 2023; Li et al., 2023b).

To address these challenges and optimize inference efficiency, various systems have been developed to save and reuse KV cache via prefix caching mechanism (Zheng et al., 2023; Kwon et al., 2023; Juravsky et al., 2024; Jin et al., 2024a; Gim et al., 2024; Gao et al., 2024). A typical workflow of these systems are demonstrate in Figure 1. These systems leverage different layers of the storage hierarchy, each with its own trade-offs. GPU memory offers the lowest latency but has the highest cost and smallest capacity, making it impractical for long-term KV cache storage. CPU memory, used by some inference engines (Zheng et al., 2023; Kwon et al., 2023), is often insufficient for large-scale online serving systems handling millions of requests per second. Consequently, local or remote disk storage has emerged as a more viable option for large-scale operations, offering a balance between cost and capacity. The trend towards disk-based KV cache storage is evident in industry practices. For example, Deepseek, a major LLM API service provider, implements prefix caching on disk, potentially reducing users' inference costs by up to 90% (Deepseek, 2024). Similarly, the multi-agent framework AutoGen employs disk-based prefix caching (AutoGen, 2024).

*Cake* aligns with the prefix caching mechanism for long-context LLM inference, improving latency by simultaneously scheduling KV computation and loading.

### 2.3 CHUNK PREFILL

Chunk prefill is a technique used to optimize the prefill stage of LLM inference, particularly for long input sequences. Unlike the decoding procedure, which is memory-bound, prefill stage is a computation-intensive process that demands significant GPU resources for extended periods. For

instance, prefilling 10,000 tokens on a 7B model on an A100 GPU takes approximately 1 seconds. Directly processing a long text prompt in its entirety for prefilling would monopolize the GPU, severely impacting the latency of other tasks.

Originally proposed in Sarathi (Agrawal et al., 2023; 2024), this method divides the input sequence into smaller, near-equal sized chunks and processes them sequentially. By breaking down large prefill requests into manageable chunks, chunk prefill allows for improved throughput and reduced prefill operations' latency impact in LLM serving systems. It enables the interleaving of prefill operations with decode operations, minimizing the blocking effect of long prompts on other tasks.This approach has been widely adopted in current mainstream LLM inference engines (Zheng et al., 2023; Kwon et al., 2023).

In vLLM's detailed implementation of chunk prefill (Kwon et al., 2023), the inference engine forms a batch of requests for each inference step based on a predetermined token budget. The scheduler prioritizes decode requests, allocating one token from the budget to each. Any remaining tokens in the budget are then assigned to prefill requests. This dynamic allocation determines the chunk size for chunk prefill operations. By giving precedence to decode requests, this approach minimizes interference with ongoing decoding requests, ensuring lower inter-token latency (ITL) for decode while efficiently utilizing available resources for prefill tasks.

*Cake* adopts chunk prefill by default as it aims for the long-context inference scenarios.

## 3 EXPLORATORY EXPERIMENTS AND MOTIVATION

Upon receiving a new prefill task, the KV cache can be obtained through various sources, including the computation on local GPU, fetching from local disk, or data streaming from a remote location, as mentioned in § 2. This section evaluates the performance of different KV cache loading or generation methods and identifies potential improvements, motivating the design of *Cake*.

**Finding 1: The latency of loading KV cache from local or remote disks is linearly correlated with the data size**.

The size of the KV cache grows linearly with the number of tokens and can be calculated as:

$$\text{KV cache size} = 4 \cdot N_l \cdot H \cdot L_{\max} \cdot B \cdot P, \tag{1}$$

Where $N_l$ is the number of layers, $H$ is the hidden size, $L_{\max}$ is the maximum context length, $B$ is the batch size, and $P$ is the precision in bytes. Using this formula, we can estimate that for the Llama3-70B model with FP16 precision, the KV cache for a single token consumes approximately 2.5 MB. Consequently, a 15-page research paper contains around 10,000 tokens, would require a great amount of memory as large as 25 GB.

Because of the size, the process of loading long-context KV cache into GPU memory is not fast because of the constrained bottleneck throughput of the I/O devices. Local storage options vary in performance: SATA SSDs offer I/O bandwidths of around 600 MB/s, HDDs are limited to about 200 MB/s, and high-end NVMe SSDs can achieve read speeds of up to 3 GB/s at a premium cost. Even in the optimal case with a 3 GB/s read bandwidth, loading the KV cache for a 10-page paper would take approximately 8 seconds. Such latency will be even longer when using remote storage, whose bottleneck becomes the network bandwidth. Typical network bandwidths rarely exceed 20 Gbps (i.e., 3GB/s) (Liu et al., 2023; Jain et al., 2023), potentially leading to transfer times of up to 10 seconds to fetch the 25 GB KV cache from a remote disk.

**Finding 2: The latency and memory usage of computing KV cache on GPUs increases quadratically with the length of the sequence.**

During the prefill stage, the computation cost for generating KV cache of later tokens (i.e., those with higher indices in the sequence) is expected to be higher than for earlier tokens This is because of the inherent nature of attention mechanisms — computing KV cache involves attention operations across the current and all preceding tokens. To test this hypothesis, we conducted experiments on long-context prefill tasks with the chunk prefill mechanism described in § 2.3. Results presented in Figure 3 demonstrate a clear pattern: the latency for each chunk linearly increases with its index; i.e., the latency for the whole sequence is quadratically correlated with the sequence length. We can also observe the KV cache memory usage linearly increases with the chunk index. The observation is critical for *Cake* to arrange computation tasks optimally.

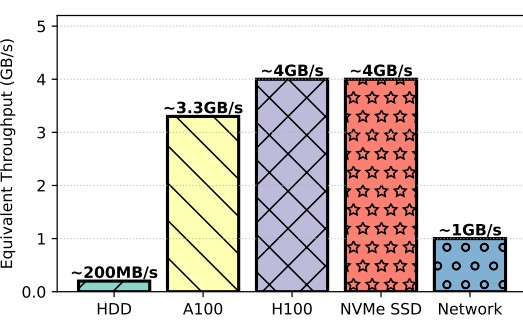

Figure 2: Equivalent KV cache loading bandwidth.

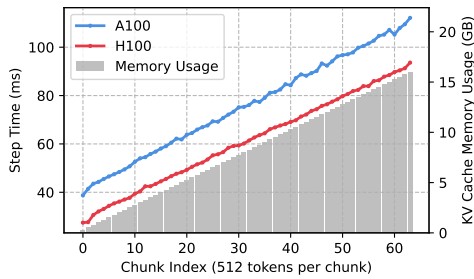

Figure 3: Prefill latency(i.e., step time)/Total Memory Usage v.s. chunk index.

**Finding 3: Computing KV cache could be even faster than loading cached KV cache**.

We compare the performance of loading and computing KV cache at various system settings. We use various GPUs to perform chunk prefill of a random context with 32k tokens through LongAlpaca-7B and -13B (Chen et al., 2023) models, implemented on vLLM (Kwon et al., 2023). We use equavalent throughput as the evaluation metric, dividing the computed or loaded KV cache file size by the time spent. We evaluate the prefill using chunk sizes as 512, as suggested by vLLM.

We present the results in Figure 2. Loading KV cache from remote disk is faster than loading from a local HDD but is slower than loading from a local SSD. Throughput of computing KV cache with 100% A100 GPU power is comparable to that of loading KV cache from SSD disk or network.

**Conclusion**. The above experiments highlighted the limitation of existing prefix caching – loading KV cache via disk or network I/O introduces substantial latency. Recomputation of KV cache, though consuming GPU power, is fast enough to potentially accelerate the prefill process. Our findings suggest that an optimal approach for prefix caching could simultaneously leverage available GPU power and I/O bandwidth. This strategy forms the core concept of *Cake*.

## 4 DESIGN OF *Cake*

We propose *Cake*, a system that adaptively utilizes both network and available computation resources to achieve low-latency KV cache loading. In this section, we first discuss the scope of *Cake* by specifying its use cases in Section 4.1. We then elaborate design details in Section 4.2 and analyze the benefit of *Cake* in Section 4.3.

### 4.1 USE CASES OF *Cake*

*Cake* works upon a LLM serving system with prefix caching as depicted in Figure 1. *Cake* is especially beneficial when the system needs to load saved KV cache from disk storage to GPU Memory for inference. It's worth noting that in cases where the required KV cache data already resides in GPU or CPU memory, it is beyond the scope of *Cake*. This is because the bandwidth between CPU and GPU is dramatically higher than disk I/O or computation speeds, then comparing with I/O, computation can provide limited help. However, given the memory size constraints on CPU and GPU and the large size of long-context caches, such scenarios are relatively infrequent, making *Cake* a crucial component for most long-context LLM inference with prefix caching tasks.

*Cake* is complementary to other optimization techniques. For instance, methods focused on reducing KV cache size (Hooper et al., 2024; Jiang et al., 2023; Kang et al., 2024) or optimizing KV cache loading from local/remote disks (Liu et al., 2023) can be used in conjunction with *Cake*, potentially yielding even greater performance improvements. Furthermore, *Cake* is designed to integrate seamlessly with state-of-the-art LLM serving systems (Kwon et al., 2023; Yao et al., 2024), enhancing their capabilities in handling long-context scenarios efficiently.

### 4.2 DESIGN DETAILS

We build *Cake* upon the chunk prefill design discussed in §2.3, where a long sequence is split into chunks, and the prefill is performed chunk by chunk. Chunk prefill is widely adopted for long-

Figure 4: Diagram illustrating the workflow of *Cake*

context LLM inference systems to offer scalable management of concurrent multiple user requests. In *Cake*, chunks serve as the fundamental unit for scheduling prefill tasks.

The key idea of *Cake* is to simultaneously leverage both local computation and I/O for data streaming to optimize the latency of KV cache prefill. For each long-context request, *Cake* determines an efficient schedule for using all available resources to accomplish KV cache prefill, i.e., determining for each chunk whether to utilize local GPU computation or data streaming from local/remote disks, as well as the sequence of these operations. As discussed in §3, the computation of KV cache on later chunks is more expensive than that on earlier chunks. Therefore, we should prioritize computation operations on chunks near the beginning of the sequence while let cache loading operations to fill later chunks. Inspired by this intuition, we design *Cake* as a bidirectional parrallelized KV cache loader as below.

As illustrated in Figure 4, upon receiving a request, *Cake* splits the sequence into chunks and initiates two simultaneous processes: (1) The local GPU computes KV cache from the beginning chunk of the prompt, progressing towards the end. (2) The data streaming process fetches KV cache starting from the last chunk, moving in reverse direction. This bidirectional approach continues until the two processes converge in the middle, signaling the completion of KV cache generation for the entire prompt. We discuss the whole algorithms in Appendix§A and the system implementation details in Appendix§B.

### 4.3 BENEFITS OF *Cake*

Our design offers several key benefits:

• **Optimized latency:** *Cake* optimally reduces the overall latency of prefill with KV cache loading by parallelizing computation and network streaming without any idle time.

• **Automatic adaptation:** *Cake* dynamically adapts to varying conditions without relying on manually defined parameters. It consistently offers latency improvements across different sequence lengths and diverse system configurations, tolerating fluctuations in network conditions and computational capabilities. This adaptability is evidenced by our comprehensive evaluation across various scenarios, as detailed in Section 5.

• **Negligible overhead:** *Cake* automatically balances the demand for computation and network resources according to dynamic situations without substantial additional overhead. Previous computation-only and network-only KV cache loading solutions (e.g., Kwon et al. (2023); Liu et al. (2023)) become special cases of *Cake* when either computation or network is unavailable. Additionally, *Cake* operates without such computation-intensive profiling phase, further reducing its operational overhead.

## 5 EVALUATION

In this section, we first introduce the setup of our experiments and then we utilize thorough experiemnts to address the following questions:

1. How does *Cake* perform under varying I/O bandwidth conditions when utilizing the full power of GPU? §5.2

2. What is the impact of context length on *Cake* performance?§5.2

3. How does *Cake* adapt to different levels of available GPU resources?§5.3

4. How effectively does *Cake* integrate with state-of-the-art KV cache compression techniques?§5.4

5. What overhead does *Cake* introduce when integrated into vLLM's inference procedure?§5.5

## 5.1 EXPERIMENT SETUP

**Models.** We evaluate *Cake* on fine-tuned long context models LongAlpaca-7B (Chen et al., 2023) based on LLama2. The per-token KV cache size is 0.5MB respectively, using 16-bit floating point as the data type.

**Evaluation Metrics.** We use time-to-first-token (TTFT) as our primary evaluation metric. TTFT is widely used in LLM inference, measuring the time between the arrival of a user query and the generation of the first token. In other words, it reflects either the time of loading stored KV cache or computing new KV cache.

**Datasets.** We evaluate *Cake* across a range of context lengths based on three datasets with different tasks: LongChat (Li et al., 2023a) for multi-turn conversation, and TriviaQA and NarrativeQA (Bai et al., 2023) for long document question-answering tasks. Based on the statistics analyzed in CacheGen (Liu et al., 2023), we found that most dataset queries fall between 5k to 16k tokens in length. Since the specific token values don't affect our evaluation of *Cake* performance (only the token length matters), we create synthetic prompts by uniformly sampling 20 data points between 5k to 16k tokens to evaluate the system's performance. To further stress test *Cake* and evaluate its performance at the upper limit of the models' capabilities, we also generate synthetic prompts with 32K tokens, which corresponds to the maximum context window supported by the LongAlpaca models.

**Baselines.** We compare *Cake* to three types of KV cache prefill/loading mechanisms:

• Compute-only methods, which employ chunk prefill to compute all the KV cache. As suggested by vLLM, the token budget size is set to 512 throughout the experiment.

• I/O Fetch-only, which loads saved KV cache from local/remote disks through Disk/Network I/O.

• KV cache Compression methods, which are orthogonal to our work. They can compress the KV cache size to make them more efficiently transferable through I/O. In our evaluation, we combine the most common 8bit quantization and CacheGen compression technique (Liu et al., 2023) with *Cake* to further evaluate its performance.

**Hardware setting.** We run our evaluation on two server configurations: 1) An NVIDIA A100 80GB GPU server equipped with a 64-core AMD EPYC 7763 CPU and 2.0TB memory. 2) An NVIDIA H100 GPU server equipped with a 26-core vCPU and 200GB memory.

**I/O Bandwidth Control.** To precisely control I/O Bandwidth with different I/O Bandwidth traces, we calculate the delay time based on the size of the chunk and network bandwidth, and then instruct LLMCache to sleep for this calculated time before fitting the data into CPU memory.

**GPU Resource Usage.** In online serving scenarios, it's common for a machine to serve multiple users' requests simultaneously. Thus, a user's prefill operation may not always have access to the full GPU resources. To evaluate different available GPU resource conditions, we utilize vLLM's token budget scheduling policy as discussed in §2.3. We schedule a partial token budget to *Cake*' prefill request and use other synthetically generated requests to consume the rest of the token budget, simulating different levels of GPU resource availability.

## 5.2 *Cake* PERFORMANCE UNDER FULL GPU RESOURCES SETTING

In this section, we evaluate *Cake* with full GPU resources and present the results in Figure 5a. We simulated three static I/O bandwidths (2000/5000/10000 Mbps) to represent different I/O conditions (HDD, network, SSD) and recorded the TTFT for a 32k token context under various settings.

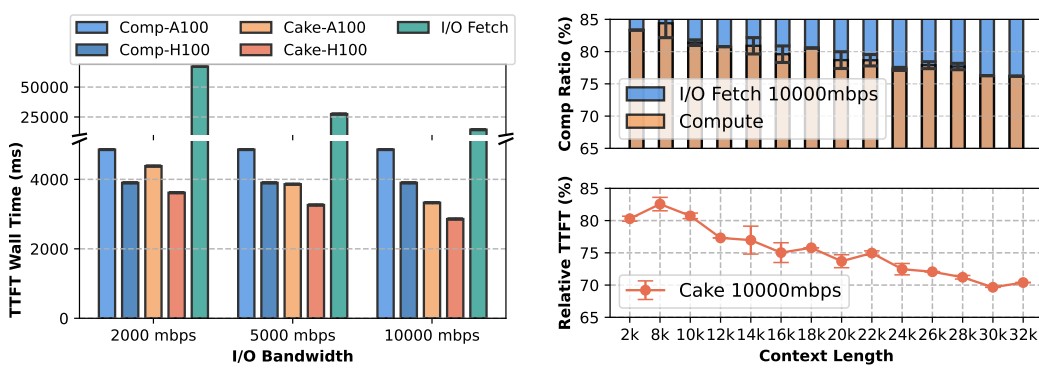

(a) Prefill 32k context of Alpaca-7b     (b) Performance of *Cake* on different context lengths

Figure 5: Evaluation of *Cake* with full GPU resources

Our observations show that compared to I/O fetch-only methods, utilizing full GPU power simultaneously reduces TTFT by 76.9-93.5% on A100 and 80.2-94.6% on H100. Moreover, compared to compute-only methods, *Cake* saves up to 31.5% on A100 and 26.7% on H100. These results demonstrate that *Cake* significantly reduces TTFT for long contexts.

We further analyzed the workload distribution between computation and fetching in *Cake* under different context lengths to understand context length effect on compute and I/O. In this setting, we fixed the compute power to use the full A100 GPU, the I/O bandwidth to 10000mbps. As shown in Figure 5b. The top figure reveals that with 10000 Mbps bandwidth, the percentage of KV cache chunks processed by computation decreases by 10% as context length increases from 2k to 32k tokens. This aligns with our observation in §2.3: as context length grows, computing KV cache for later tokens becomes more time-consuming, while I/O time remains constant. Consequently, at the convergence point, a smaller percentage of tokens will be processed by computation as context length increases.

The bottom figure illustrates the ratio of *Cake* TTFT to compute-only TTFT. We observe that as context length increases from 2k to 32k tokens, the relative TTFT is reduced by 15%. This demonstrates that *Cake* becomes even more efficient compared to compute-only methods as context length increases.

Referring to Figure 3, we note that computing later chunks also requires more GPU memory. KV cache memory consumption grows linearly with token size. With *Cake* under 10000 Mbps bandwidth, only 75% tokens need to do computation and store KV cache in GPU memory, while the saved 25% space can be allocated for other short requests. This further highlights *Cake*'s memory efficiency.

### 5.3 *Cake* Performance under Different GPU Usage

In this section, we evaluate *Cake* under different GPU resource usage scenarios to assess its performance in real-world settings where GPU resources need to be shared among multiple users. The results are demonstrated in Figure 8. We control GPU usage by scheduling limited token budgets as described in §5.1. We present two representative results with a context length of 14k tokens, with additional results available in Appendix§C.

Compared to the compute-only method, *Cake* leverages I/O prefetching to reduce latency by 5.3-68.1%. Higher I/O bandwidth allows *Cake* to provide greater benefits. When compared to I/O-only methods, *Cake* utilizes computation to reduce latency by 27.4-93.7%, with higher computation power yielding more significant improvements.

Specifically, using the compute-only method, reducing computation power from 90% to 10% results in an 8.22x increase in prefilling latency, leading to an additional 14.34s delay. However, *Cake* automatically fetches more chunks of KV cache via I/O, reducing latency by up to 68.1% compared to the compute-only method and 27.4-65.1% compared to the I/O-only method under the same condi-

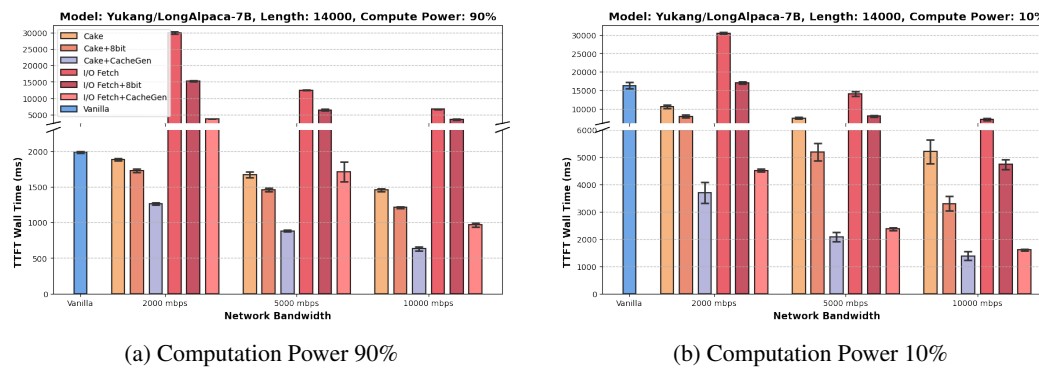

(a) Computation Power 90%     (b) Computation Power 10%

Figure 6: Comparing *Cake* with compute-only and different I/O fetching methods.

tions. This observation generalizes across different context lengths and compute powers, as shown in our results in Appendix§C, demonstrating that *Cake* consistently achieves the fastest prefilling speed compared to both computation-only and I/O-only methods.

In conclusion, *Cake* significantly reduces prefilling latency when available computation power is insufficient by leveraging KV cache fetching. This feature can be exploited to utilize fragmented computational resources for prefilling long context requests without adversely affecting other users' experience.

### 5.4 INCORPORATING KV CACHE COMPRESSION WITH *Cake*

In this section, we show that *Cake* can incorporate with state-of-the-art KV cache compression technologies to further boost its performance. We use the same settings as in §5.3 and apply the widely used 8bit-quantization and the latest CacheGen (Liu et al., 2023) to reduce the size of KV cache. Theoretically, they will reduce the fetching time by 2 and 8.6 respectively, thus lowering the bandwidth requirement.

Our results in Figure 8 demonstrate that 8bit-quantization and CacheGen significantly enhance I/O bandwidth, achieving speedups of up to 1.95x and 7.97x respectively. Leveraging these improvements, *Cake* is able to fetch more KV cache chunks during computation, resulting in a remarkable 11.83x speedup compared to computation with only 10% GPU power. Furthermore, *Cake* combined with CacheGen reduces time-to-first-token (TTFT) by 32.8-73.5% compared to *Cake* with raw KV cache, showcasing its ability to efficiently utilize all available resources to minimize latency.

Conversely, when *Cake* has access to 90% of GPU resources, it outperforms CacheGen-only fetching by reducing TTFT by 66.4%. This demonstrates *Cake*'s adaptability across different resource availability scenarios. As detailed in Appendix§C, *Cake* consistently achieves superior performance compared to both computation-only and fetching-only methods across a wide range of scenarios, highlighting its versatility and effectiveness in optimizing prefilling latency.

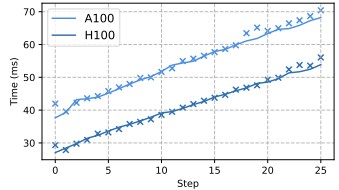

Figure 7: Per-step inference time in vLLM before and after integration with *Cake*. The solid line represents the step time without *Cake*, while the 'x' markers indicate step times with *Cake*.

### 5.5 OVERHEADS OF *Cake* IN LLM SERVING SYSTEM

To demonstrate *Cake* has negaligible overheads, we compare the duration of each engine step on original vLLM and vLLM with *Cake*. As is shown on Figure 7, we launch a chunk prefill job on A100 and H100 server, and the chunk prefill time of vLLM with *Cake* basically follows the trace of original vLLM. This results prove that *Cake* introduce negaligible overheads, as it only needs to check whether the next chunk is already fetched and doesn't have to schedule anything.

## 6 DISCUSSION

In this section, we discuss the benefits and potential overheads introduced by *Cake*. Our system effectively utilizes both compute and I/O resources to reduce KV cache loading latency, thereby improving the Time To First Token (TTFT).

**Comparison with Compute-Only Methods.** Unlike approaches that rely solely on computation, *Cake* leverages I/O bandwidth to reduce the computational load, resulting in lower latency. This is particularly advantageous as I/O resources are often less costly and more readily available compared to high-performance compute resources.

**Comparison with I/O-Only Methods.** While *Cake* primarily focuses on efficient I/O utilization, it also employs compute resources to further reduce TTFT. This dual approach may introduce some additional computational overhead. However, as discussed in §5.3 and §2.3, we can strategically use only the unused token budget for chunk prefill computation. This approach minimizes the system's computational overhead due to the batching effect and efficient GPU utilization. The added computation is efficiently processed alongside other tasks, leveraging GPU parallelism. Even if there are no unused token left, we can pause the compute procedure and let I/O contribute more to the procedure.

**Cost-Benefit Analysis.** In many online services, users who pay more often receive higher priority and faster request processing. *Cake* is particularly well-suited for these scenarios, offering significantly reduced TTFT for a marginal increase in cost.

## 7 CONCLUSION

In this paper, we introduced *Cake*, a novel approach that efficiently utilizes both I/O and compute resources to reduce Time To First Token (TTFT) for LLM serving systems with prefix caching. *Cake* dynamically adapts to varying resource conditions, seamlessly integrating with existing KV cache optimization techniques to achieve optimal latency with minimal overhead. Our evaluation shows *Cake* outperforms both compute-only and I/O-only methods, reducing TTFT by up to 95% compared to baselines. *Cake* balances I/O and compute resources to maximize performance gains without significantly increasing costs. As a simple plug-in solution, *Cake* is easily implementable in existing LLM serving systems with prefix caching, offering substantial performance improvements and straightforward integration to enhance the responsiveness and efficiency of LLM services.

## 8 ETHICS STATEMENT

Our research on *Cake*, a system for reducing Time To First Token (TTFT) in LLM serving systems, primarily focuses on improving computational efficiency without directly involving human subjects or sensitive personal data. However, we acknowledge broader ethical implications: potential disparities in service quality based on user access levels, the positive environmental impact through optimized resource utilization, our commitment to transparency and reproducibility, the risk of misuse in scaling harmful LLM applications, and the importance of adhering to ethical AI deployment guidelines. We declare no conflicts of interest and affirm our commitment to the ICLR Code of Ethics, having conducted this research with integrity and in compliance with established ethical standards in AI and computer science. While *Cake* aims to enhance LLM services, we encourage implementers to consider fair allocation strategies, implement safeguards against misuse, and continue efforts to minimize the environmental footprint of AI systems.

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

# A  DETAILS OF *Cake* ALGORITHM

The workflow of *Cake* can be described in detail as follows:

1. Upon receiving a request, *Cake* first splits the input sequence into chunks of a predetermined size (e.g., 512 tokens as suggested by vLLM).

2. Two pointers are initialized: $compute\_ptr$ starting at the beginning of the sequence (index 0), and $io\_ptr$ at the end of the sequence (index $total\_tokens - 1$).

3. Two parallel processes are initiated: a) A GPU computation thread that starts from $compute\_ptr$ and moves forward. b) An I/O streaming thread that starts from $io\_ptr$ and moves backward.

4. The GPU computation thread: - Computes KV cache for chunks starting from $compute\_ptr$. - After each chunk computation, it updates $compute\_ptr$ by adding the chunk size. - Continues until $compute\_ptr$ reaches or surpasses $io\_ptr$, or until the required KV cache is found in CPU memory.

5. The I/O streaming thread: - Fetches pre-computed KV cache for chunks ending at $io\_ptr$ from storage (local or remote) to CPU memory. - After each chunk fetch, it updates $io\_ptr$ by subtracting the chunk size. - Continues until $io\_ptr$ reaches or goes below $compute\_ptr$.

6. The process concludes when the two pointers meet or cross each other, indicating that KV cache for the entire sequence has been either computed or loaded.

7. Finally, *Cake* returns the complete KV cache for the entire sequence, ready for use in the subsequent inference steps.

This bidirectional approach allows *Cake* to efficiently utilize both computational and I/O resources simultaneously, minimizing idle time and optimizing the overall latency of KV cache preparation for long-context LLM inference.

---

**Algorithm 1** *Cake* Bidirectional KV cache Loading Algorithm

---

1: **procedure** COMPUTEKV
2:     **while** $compute\_ptr < io\_ptr$ **do**
3:         **if** ISINCPUMEMORY($compute\_ptr$, $CHUNK\_SIZE$) **then**
4:             Signal I/O worker to stop
5:             **break**
6:         Compute KV cache for chunk starting at $compute\_ptr$ using GPU
7:         $compute\_ptr \leftarrow compute\_ptr + CHUNK\_SIZE$
8: **procedure** FETCHKV
9:     **while** $compute\_ptr < io\_ptr$ **do**
10:        Fetch KV cache for chunk ending at $io\_ptr$ from storage to CPU Memory
11:        $io\_ptr \leftarrow io\_ptr - CHUNK\_SIZE$
12: Initialize CPU Memory, $compute\_ptr = 0$, $io\_ptr = total\_tokens - 1$
13: Start COMPUTEKV in a new thread
14: Start FETCHKV in a new thread
15: Wait for both threads to complete
16: **return** KV cache for the entire sequence

---

# B  IMPLEMENTATION

We implement *Cake* by extending LMCache (LMCache, 2024) and integrating it with vLLM (Kwon et al., 2023), adding approximately 1,000 lines of code.

## B.1  ENHANCEMENTS TO LMCACHE

LMCache, originally developed as the KV cache management backend for CacheGen (Liu et al., 2023), hashes token chunks into keys for efficient KV cache retrieval. To enable *Cake* to continuously receive KV cache in the background, we introduce the following enhancements:

**Asynchronous Retrieval**  We implement an asynchronous get operation to complement LM-Cache's existing asynchronous put functionality. This involves creating a dedicated worker thread

that continuously reads chunk keys from a task queue and retrieves the corresponding KV cache to memory. Upon successful retrieval, the chunk's key is added to a resident dictionary for quick access.

**Buffer Preallocation**   We modify LMCache to preallocate chunk buffers as soon as a chunk key is pushed to the queue. This optimization allows the worker to immediately write received KV cache into memory and proceed to the next chunk without delay.

### B.2   INTEGRATION WITH LLM SERVING SYSTEMS

*Cake* operates concurrently with LLM serving systems like vLLM. The integration process works as follows:

1. Upon receiving a request, *Cake* divides it into chunks based on the scheduled token budget.

2. Hashed keys for these chunks are pushed to the task queue in reverse order using the *push_seq* API.

3. While the asynchronous get worker fetches KV cache from the end of the sequence, vLLM begins chunk prefill from the start.

4. After each vLLM engine step, *Cake* checks if the next chunk of tokens is already in the resident dictionary using the *is_resident* API.

5. If the chunk is resident, *Cake* interrupts the chunk prefill process and directs vLLM to begin token generation.

6. If the chunk is not resident, chunk prefill continues until it encounters a chunk present in the dictionary.

This bidirectional approach allows *Cake* to efficiently utilize both I/O and computational resources, potentially reducing the Time To First Token (TTFT) for long-context LLM inference tasks.

## C   PERFORMANCE OF *Cake* UNDER DIFFERENT CONDITIONS

We compare the performance of *Cake* with compute-only and I/O-only methods. We also incorporate 8bits-quantization and CacheGen into *Cake*, which further boost our performance. Selected diagrams includes 9k and 14k context lengths with 10-50-90% of GPU resources. Bandwidth we test ranges from 2000-10000 mbps. Across all the conditions, *Cake* with CacheGen produce the best performance with the lowest TTFT.

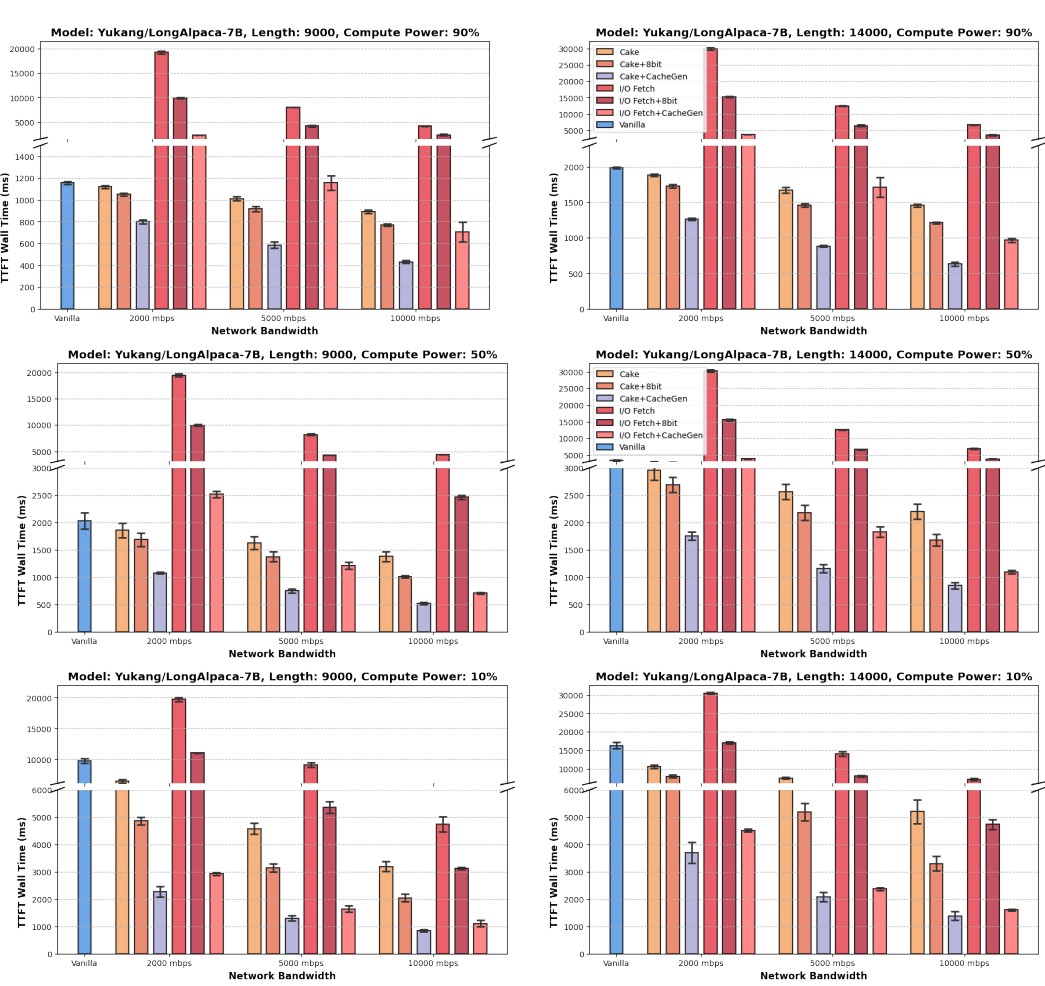

Figure 8: Comparing *Cake* with recompute only and different I/O fetching methods.

