# OpenReview forum: "Compute Or Load KV Cache? Why Not Both?"
_ICLR.cc/2025/Conference — ICLR 2025 Conference Withdrawn Submission_

### Official Review · Reviewer_bNCp · 2024-10-28

**Soundness:** 2
**Presentation:** 2
**Contribution:** 2
**Rating:** 3
**Confidence:** 4

**Summary:**

The paper presents Cake, a KV cache loader designed to optimize the latency of long-context LLM inference by balancing GPU computation and I/O throughput. Traditional prefix caching methods, though effective in reusing KV cache to reduce computation, suffer from high latency due to I/O bottlenecks when loading cache from disks. Cake addresses this challenge through a bidirectional parallelized KV cache generation strategy: it computes the KV cache sequentially from the start of the token sequence on the GPU while simultaneously loading the cache in reverse from disk, maximizing resource utilization. Cake also supports dynamically adapting to system conditions without manual tuning, ensuring optimal performance across varying workloads and architectures. This approach improves TTFT performance by 68.1% compared to compute-only methods and 94.6% compared to I/O-only solutions.

**Strengths:**

++ The paper targets an important problem in LLMs -- the balance of KV-transfer and KV-recomputation

**Weaknesses:**

-- The motivation is not clear

-- Lacks of design details

-- The evaluation scenarios are not clear

Please see the details in the Questions.

**Questions:**

**Motivation:**

The main design advantage of Cake depends on the effective reuse of the KV cache. However, the paper does not provide empirical data to demonstrate the extent of KV cache reuse. Without supporting evidence on the percentage of KV cache reuse, it is difficult to assess the potential impact and effectiveness of the proposed solution.



**Design details:**

How does Cake determine whether a newly requested KV cache is identical to an existing one? Given that the system employs a chunk-based approach, matching all tokens requires multiple memory accesses. How does Cake efficiently identify if the KV cache for a specific token sequence already exists in the cache?

What mechanisms does Cake use to locate where a specific KV cache is stored, whether on CPU memory or SSD? Does Cake perform searches across all CPU and SSD storage locations? If so, how does the system manage the latency introduced by these extensive memory accesses?

How does Cake manage KV caches within its streaming pipeline? Specifically, are all previously generated KV caches stored exclusively on either SSDs or CPU memory or is there a hierarchical storage strategy?
 Are all previously generated KV caches stored on either the SSD or CPU memory? Are there any eviction for KV caches stored on SSD or CPU memory? If eviction occurs, what replacement policy does the system use (e.g., Least Recently Used, Least Frequently Used)? During the bidirectional loading process, how does Cake handle scenarios where a requested KV cache is not found in either CPU memory or SSD storage?

Given that position information is encoded within the KV cache, how does Cake ensure that reused KV caches maintain accurate position encoding for new requests? When a new request leverages a previously stored KV cache, how does the system verify and adjust the position information to align correctly with the current token sequence?


**Evaluation:**

It would be better to provide the hit rate of different storage solutions to better understand the performance gains.

The evaluation includes multi-turn conversations, but it is unclear whether Cake captures reuse within multiple turns of a single conversation or across different conversations. For long-context tasks, I am not clear where the reuse comes from. Is it derived from repeated access to specific segments of the document or overlapping contexts?

---

### Official Review · Reviewer_11qw · 2024-11-03

**Soundness:** 2
**Presentation:** 2
**Contribution:** 1
**Rating:** 3
**Confidence:** 5

**Summary:**

This paper discusses the system trade-off between reusing cached KV cache or dynamically recomputing it during the LLM generative inference computation. In addition to discussing the trade-offs between reusing cached key-value (KV) cache and dynamically recomputing it, this paper delves into the hybrid approach implemented in a system named Cake. By using bidirectional parallelism, Cake dynamically adapts its KV cache loading strategy to balance between computational and I/O demands, leveraging both GPU and storage resources. The system autonomously optimizes for different hardware configurations and workload requirements. Empirical experiments has been conducted to evaluate the effectiveness of the proposed method.

**Strengths:**

- S1. Understanding the system cost of the LLM generative inference workflow is an essential step to building an efficient LLM inference serving system. The paper considers an interesting aspect of the trade-off between reusing cached KV cache and dynamically recomputation.

**Weaknesses:**

- W1. The proposed approach lacks enough novelty. This paper fails to discuss its unique contribution from some recent works that have explored similar system efforts, for example:
  - (i) XUANLEI, ZHAO, et al. "HeteGen: Efficient Heterogeneous Parallel Inference for Large Language Models on Resource-Constrained Devices." Proceedings of Machine Learning and Systems 6 (2024): 162-172.
  - (ii) Park, Daon, and Bernhard Egger. "Improving Throughput-oriented LLM Inference with CPU Computations." Proceedings of the 2024 International Conference on Parallel Architectures and Compilation Techniques. 2024.

- W2. The imprecise formulation in Section 3. For example, in Finding 1, the KV cache size is estimated by $KV cache size = 4·N_l·H·L_{max}·B·P$, this fails to consider the widely used modification of the transformer architecture with group-query attention (GQA), which can reduce the memory footprint by a factor of 4, 8 or more, such a factor could lead to significant shit of the trade-off discussed in this paper.

- W3. The experimental evaluation is poorly designed -- the scenario is far from the state-of-the-art infrastructure to serve generative LLM inferences. The state-of-the-art real system to support generative inference could leverage a fast Infiniti-band or RoCE network to share KV-cache, a couple of magnitudes faster than the simulated latency injected in the experiments.

**Questions:**

Please address the corresponding concerns listed in the Weaknesses Section.

**Details Of Ethics Concerns:**

Not Applicable.

---

### Official Review · Reviewer_vvnQ · 2024-11-03

**Soundness:** 4
**Presentation:** 4
**Contribution:** 3
**Rating:** 6
**Confidence:** 4

**Summary:**

This paper introduces a novel approach, Cake, for KV cache generation during the pre-fill stage in LLM inference. Cake optimizes KV cache generation by exploiting both – the available, free compute resources on the GPU and the I/O bandwidth to load the KV cache from external/remote disks. The authors conduct exhaustive experiments to verify their hypothesis – KV cache generation latency during the pre-fill stage can be significantly reduced by combining KV cache computation on GPU with KV cache loading from memory/disk. The proposed bidirectionally parallel KV cache generation strategy during pre-fill outperforms both computation-only and load-only baselines. Further, Cake integrates seamlessly with chunk-prefill, cache compression and token budget scheduling policy (vLLM) methods, used to optimize LLM inference, underscoring its adaptability and generality.

**Strengths:**

Clarity: The paper is clearly written, organized well and structured cohesively.

Motivation: The motivation behind the paper is clear – the authors define their problem statement guided by exploratory experiments. These experiments suggest that current KV cache generation methods can be optimized by combining both – computation and loading.

Simplicity: The proposed method, Cake, is simple and effective. The bidirectional KV cache generation strategy proposed in Cake exploits the available compute resources on GPU and the I/O bandwidth to simultaneously compute and load the KV cache. Further, the authors integrate Cake with vLLM to demonstrate its efficacy under situations where limited GPU resources are available and multi-user requests need to be served with minimal interruption.

Efficacy: Cake is effective at reducing the overall latency for time-to-first-token (TTFT), across a wide variety of situations. The authors verify their claims via clear and concise experiments that highlight Cake’s efficacy.

Practicality: Cake is a simple and elegant method – making it practical to be implemented in real time systems serving LLM inference user queries.

**Weaknesses:**

Typos: There are several minor typos (ex. Line 036, Sonnet misspelled as Sounet). The authors should rectify these typos.

Scalability: The authors conduct experiments, across varying situations, on only a single LLM (LongAlpaca-7B). This raises concerns regarding the scalability of the proposed method across larger LLMs.

Limited multi-user experiments: To assess the multi-user, real-time performance of Cake, the authors conduct experiments across static variations of GPU resources available and network bandwidth. Practical LLM inference serving systems, with multi-user queries, generally have a dynamic variation in both the available GPU resources and the network bandwidth. This raises concerns regarding efficacy of Cake in real-time, dynamically varying LLM inference serving systems.

**Questions:**

How well does Cake perform across other LLMs, larger and smaller than LongAlpaca-7B? Does the performance scale well for larger LLMs?

How well does Cake perform under a dynamically varying system available GPU resources and network I/O bandwidth? Under such a dynamic setting (true for real-time LLM inference serving systems), how does Cake schedule compute vs load for KV cache generation?

Can the authors include experiments associated with such dynamic workloads? Without such experiments, the claim of “automatic adaptation” made by the paper cannot be substantiated. The addition of such experiments will significantly strengthen the paper and its claims.

---

### Official Review · Reviewer_KPZk · 2024-11-04

**Soundness:** 2
**Presentation:** 3
**Contribution:** 2
**Rating:** 5
**Confidence:** 2

**Summary:**

The authors introduce a new framework called Cake, which parallelizes the computation and loading of KV caches. A key observation is the absence of a dynamic KV cache management mechanism capable of optimizing LLM inference latency across varying system conditions. By employing a bidirectional parallel strategy, Cake can simultaneously execute computations and load data by leveraging available resources, thereby significantly reducing overall latency. The paper also discusses the design of Cake, which adapts to diverse scenarios in terms of system architecture and available resources, all without requiring manual parameter tuning. In conclusion, this work presents an innovative approach to optimally balance the utilization of GPU power and I/O bandwidth, achieving minimal latency for long-context LLM inference.

**Strengths:**

* Striking a balance between the utilization of GPU power and I/O bandwidth for optimizing LLM inference latency is an interesting idea. This approach effectively utilizes both computational resources and I/O resources, making it an intuitive and efficient solution.
* The paper has good coherence, and is well-structured. The background section explains the related work in an easy understandable way.
* The paper is also very clear with thorough experiments and analysis.

**Weaknesses:**

* The observation regarding Cake is based on the assumption that “loading KV cache from a remote disk is faster than loading from a local HDD but slower than loading from a local SSD” (Lines 236-237). This point is somewhat confusing, as we believe that loading from a remote disk would require data to be copied from the disk to memory and then transmitted over the network, which could significantly impact bandwidth. We request clarification on this issue.
* Additionally, the experimental setup does not include the network configuration, nor does it present results from testing with additional remote disks. We suggest that the authors discuss the broader applicability and limitations of this work, specifically outlining the conditions under which it is effective and where it may fall short.
* Furthermore, the experiment setup doesn’t incoorparate the network configuration, and include the results when adding more remote disks. It is suggested that the authors discuss the broader applicability and limitations of this work, specifically under what conditions it works and when it may not be effective.
* There is a lack of analysis regarding the quality of the generated KV cache. It remains unclear whether the quality of the cache is compromised when employing both compute and load strategies simultaneously. For example, by introducing Cake optimization, will the generated KV cache differ from those produced using only the compute or load methods?
* There is a lack of time complexity analysis for Cake.
* From the evaluation results, there still lacks of evaluation across different model scales. The paper primarily evaluates the LongAlpaca-7B and LongAlpaca-13B models. While these models are representative, the study does not include larger-scale models, such as Llama3-70B with 70 billion parameters, or models with different architectures. Variations in model scale and architecture could potentially impact the performance of Cake.
* The second major concern about the evaluation pertains to the choice of comparison baselines. The baseline methods, especially the specific implementation details of the "computation only" and "I/O retrieval only" approaches, are not clearly outlined. Different implementations can significantly affect performance. So, If these baselines were not optimized to their fullest potential, it could lead to an underestimation of the performance of existing technologies. Please clarify the optimal configurations of these baselines.
* There is no discussion of the limitations of Cake.

**Questions:**

* Striking a balance between the utilization of GPU power and I/O bandwidth for optimizing LLM inference latency is an interesting idea. This approach effectively utilizes both computational resources and I/O resources, making it an intuitive and efficient solution.
* The paper has good coherence, and is well-structured. The background section explains the related work in an easy understandable way.
* The paper is also very clear with thorough experiments and analysis.

**Details Of Ethics Concerns:**

None.

---

### Note · Authors · 2024-11-24

**Comment:**

We would like to express our sincere gratitude to all reviewers for their valuable and insightful feedback. After careful consideration, we have decided to withdraw our submission and proceed to the future venue, which provides us with more opportunities to better address comments provided by the reviewers.

**Withdrawal Confirmation:**

I have read and agree with the venue's withdrawal policy on behalf of myself and my co-authors.